# Direct Testing for KPC-Mediated Carbapenem Resistance from Blood Samples Using a T2 Magnetic Resonance Based Assay

**DOI:** 10.3390/antibiotics10080950

**Published:** 2021-08-06

**Authors:** Giulia De Angelis, Riccardo Paggi, Thomas J. Lowery, Jessica L. Snyder, Giulia Menchinelli, Maurizio Sanguinetti, Brunella Posteraro, Antonella Mencacci

**Affiliations:** 1Dipartimento di Scienze Biotecnologiche di Base, Cliniche Intensivologiche e Perioperatorie, Università Cattolica del Sacro Cuore, 00168 Roma, Italy; giulia.deangelis78@gmail.com (G.D.A.); giulia.menchinelli@unicatt.it (G.M.); brunella.posteraro@unicatt.it (B.P.); 2Dipartimento di Scienze di Laboratorio e Infettivologiche, Fondazione Policlinico Universitario A. Gemelli IRCCS, 00168 Roma, Italy; 3Microbiologia Medica, Dipartimento di Medicina, Università degli Studi di Perugia, 06129 Perugia, Italy; paggi.riccardo@gmail.com (R.P.); antonella.mencacci@unipg.it (A.M.); 4T2Biosystems, Inc., Lexington, Boston, MA 02421, USA; scientificaffairs@t2biosystems.com (T.J.L.); jsnyder@t2biosystems.com (J.L.S.); 5Dipartimento di Scienze Mediche e Chirurgiche, Fondazione Policlinico Universitario A. Gemelli IRCCS, 00168 Roma, Italy; 6Dipartimento di Diagnostica per Immagini e di Laboratorio, Ospedale Santa Maria della Misericordia, 06129 Perugia, Italy

**Keywords:** antimicrobial resistance, blood sample, bloodstream infection, direct detection, KPC carbapenemase, magnetic resonance, T2Resistance panel

## Abstract

Molecular-based carbapenem resistance testing in Gram-negative bacterial bloodstream infections (BSIs) is currently limited because of the reliance on positive blood culture (BC) samples. The T2Resistance™ panel may now allow the detection of carbapenemase- and other β-lactamase encoding genes directly from blood samples. We detected carbapenem resistance genes in 11 (84.6%) of 13 samples from patients with BC-documented BSIs (10 caused by KPC-producing *Klebsiella*
*pneumoniae* and 1 caused by VIM/CMY-producing *Citrobacter freundii*). Two samples that tested negative for carbapenem resistance genes were from patients with BC-documented BSIs caused by KPC-producing *K. pneumoniae* who were receiving effective antibiotic therapy. In conclusion, our findings suggest that the T2Resistance™ panel can be a reliable tool for diagnosing carbapenem-resistant Gram-negative bacterial BSIs.

## 1. Introduction

Bacterial bloodstream infection (BSI) continues to represent a major public health concern [1], particularly in the case of infection-associated sepsis [2], leading to high morbidity and mortality rates around the world [3]. Prompt initiation of effective antibiotics is crucial to limit unfavorable outcomes [4], and it is unsurprising that the sepsis management bundle has included timely antimicrobial therapy administration and diagnostic testing as key components [5]. Rapid identification of the causative pathogen allows time to diagnosis and targeted therapy of BSI/sepsis to be considerably shortened [6]. Since blood culture (BC) remains the reference standard in BSI diagnostics [7], positive BC testing in the clinical microbiology laboratory has embraced new panel-based molecular assays [8]. In addition to identifying the most common pathogens isolated from BCs (e.g., *Enterobacterales*) within few hours, these assays can detect antimicrobial resistance determinants, such as the *Klebsiella pneumoniae* carbapenemase (KPC), which is encoded by the *bla*_KPC_ gene [9]. The KPC has recently been identified as a major determinant of carbapenem resistance among Gram-negative bloodstream isolates from European (e.g., Italy, Greece, etc.) or non-European (e.g., USA, Brazil, etc.) countries [1,10,11] However, other carbapenemase (e.g., NDM) or β-lactamase (e.g., AmpC) encoding genes are increasingly recognized as determinants capable of rendering Gram-negative organisms resistant to multiple antimicrobial drugs [12]. Especially with multidrug-resistant Gram-negative BSIs [13], the molecular assays’ turnaround time—if compared to the phenotypic or genotypic antimicrobial resistance detection methods performed on the bacterial isolates grown from BCs [14]—would allow guiding the therapeutic decisions some days before culture-based antimicrobial susceptibility testing (AST) results are available [6]. Due to their reliance on positive BCs, these assays lead so far to delays in appropriate antimicrobial therapy [13] that may translate to increases in Gram-negative BSI associated morbidity and mortality [15].

Based on the T2 Magnetic Resonance (T2MR^®^) technology (T2Biosystems, Lexington, MA, USA), which is able to detect signals by amplicon-induced agglomeration of superparamagnetic particles [16], the T2Resistance™ RUO (research use only) panel has been developed to detect the carbapenemase (KPC, OXA-48, NDM, VIM, IMP) and AmpC (CMY, DHA) encoding genes associated with BSI-causing Gram-negative bacteria directly in whole blood (https://www.t2biosystems.com/products-technology-ous/pipeline-ous/t2resistance-panel-ous/, accessed on 2 August 2021). Eliminating the need for culturing BSI pathogens, this assay uses the same T2Dx instrument as the two FDA-cleared and CE-marked direct-from-blood detection panels, which were specifically designed to identify multiple bacteria (T2Bacteria^®^ panel) and *Candida* (T2Candida^®^ panel) causing BSI [16]. In practice, T2Dx automatically concentrates and lyses microbial cells from the patient’s blood—contained in a standard K_2_ EDTA Vacutainer collection tube—before DNA is amplified by PCR and target-specific primers and then, amplified products (also termed amplicons) are detected. The development of the T2Resistance™ RUO panel was aimed at circumventing the inherent limitations of both T2Bacteria^®^ and T2Candida^®^ panels [13], but this heralded the need for studies evaluating the T2Resistance™ RUO panel.

In this article, we report on a five-month experiment with the T2Resistance™ RUO panel conducted in the clinical microbiology laboratories of two tertiary care University hospitals. Both laboratories serve hospitals located in Italy, which is considered an endemic area for multidrug-resistant Gram-negative bacteria.

## 2. Results

We studied 13 BSIs caused by carbapenemase-producing Gram-negative bacteria, 12 of which were *Klebsiella pneumoniae* and 1 was *Citrobacter freundi*. T2Resistance™ panel allowed to detect carbapenemase (10 KPC and 1 VIM) or AmpC (1 CMY) genes, respectively, in whole blood samples from 11 patients with positive BCs for KPC-producing *K. pneumoniae* (*n* = 10) or VIM/CMY-producing *C. freundii* (*n* = 1). Conversely, T2Resistance™ panel failed to detect carbapenemase (KPC) genes in whole blood samples from 2 patients with positive BCs for KPC-producing *K. pneumoniae*. This resulted in a positive percent agreement between T2Resistance™ panel results and BC results of 85.7 (95% confidence interval [CI], 60.1–96.0; 12/14).

As shown in Figure 1, according to the time from positive BC result, samples obtained on day 1 (*n* = 8) had positive T2Resistance™ panel results that agreed with BC results for 87.5% (7/8). Samples obtained on day 2 (*n* = 4) had positive T2Resistance™ panel results that agreed with BC results for 75.0% (3/4). One of four samples had also a positive T2Resistance™ panel result for the CMY gene (which was confirmed by PCR-sequencing analysis). The sample obtained on day 3 had a positive T2Resistance™ panel result that agreed with the BC result.

The two samples that tested false negative with the T2Resistance™ panel were from patients who were receiving effective antibiotic therapy (i.e., ceftazidime/avibactam) from 10 h and 40 h before sampling, respectively. Unlike others, these samples were drawn within 2 h after the therapeutic antibiotic dose, likely when the antibiotic’s bacterial killing activity was maximal. In the other patients, sampling occurred at 6−8 h of the receipt of aforementioned dose.

## 3. Discussion

To the best of our knowledge, this is the first report of T2Resistance™ panel data on whole blood samples from patients with microbiologically documented bacterial BSI, mostly caused by *K. pneumoniae*. We were able to detect the KPC-carbapenemase gene in a high percentage of patients sampled, and treated with potentially active antibiotics, within at least 36 h after a positive BC result was available. Two patients with KPC-producing *K. pneumoniae* BSI had a negative result with the T2Resistance™ panel while they were receiving antibiotics that might have resulted in the absence of (nonviable) organisms in the blood. However, delays between BC and T2Resistance™ panel sampling—the nature of the study did not imply that BCs were concomitantly drawn to the samples tested with the T2Resistance™ panel—made it difficult to show that the two (false) negative results were actually (BC-) negative results. Consequently (and consistent with the small sample number in this study), we did not assess sensitivity, specificity and positive and negative predictive values for the T2Resistance™ panel.

Despite being conceived as a “companion test”, we did not use the T2Bacteria^®^ panel—which detects five different bacterial species causing >50% of all BSIs and displaying resistance to multiple antibiotics [17]—simultaneously with the T2Resistance™ panel. In our previous single-center study, which tested 140 samples from 129 patients with BSI, a research prototype of the T2Bacteria^®^ panel showed 83.3% sensitivity and 97.6% specificity for proven BSI caused by target bacteria [18]. Notably, the sensitivity improved to 89.5% when “true infection” criteria (e.g., same bacterium isolated from another site) were used to interpret results [18]. Later, the commercially available, FDA-cleared version of the T2Bacteria^®^ panel was evaluated in a multicenter study with 1427 patients who had paired BC and T2Bacteria testing [19]. The per patient sensitivity and specificity of the T2Bacteria^®^ panel for proven BSI were both 90%, and the specificity improved to 96% when assuming both probable and possible BSIs to be true positives that were missed by BCs. Unlike the T2Bacteria^®^ panel, at the time of present writing, no data from independent studies were available for comparison purposes. Nonetheless, mirroring the previous experience with the T2Bacteria^®^ panel, our findings with the T2Resistance™ panel are expected to be confirmed in the near future.

Culture-independent molecular methods emerged several years ago to offer more rapid results while allowing the determination of the underlying genetic basis of antimicrobial resistance [20]. However, important factors—including costs and laboratory personnel staffing—need to be considered when implementing rapid methods in laboratory diagnostic workflows [13]. Among carbapenemase-producing *Enterobacterales*, *K. pneumoniae* is a global public health menace, with KPC-producing *K. pneumoniae* being the most prevalent Gram-negative species identified in Italy and some other countries [21]. Novel agents such as ceftazidime/avibactam, meropenem/vaborbactam, and imipenem/cilastatin/relebactam are practical options to combat carbapenem-resistant infections [21]. Thus, rapid molecular tests—usually designed to identify a limited spectrum of microbial species and of antimicrobial resistance determinants—may be reasonably considered for guiding therapeutic decisions in settings where carbapenemases are the major driver of carbapenem resistance [13]. Otherwise, in settings without local prevalence of carbapenemase genes, these tests may have particular value if used in conjunction with the currently available phenotypic methods for either microbial identification or AST—both applied to organisms from BCs after amplification by subculture. We recall that the time to results with the T2Resistance™ panel is 3−5 h, which is 24−48 h quicker than the results from routinely performed AST. However, acquiring such a molecular test—the expected cost is 140.00 EUR per sample—in the clinical microbiology laboratory would be expensive if all patients with possible BSIs are routinely screened using the T2Resistance™ panel.

If the T2Resistance™ panel has the potential to be the sole test or one of tests by which the antimicrobial resistance gene detection may be performed in a timely manner is unknown to date. Reviewing data from studies dealing with nucleic acid amplification-based methods, i.e., rapid tests for both identification and antimicrobial resistance gene detection of Gram-negative bacteria directly from whole blood [13], it is noteworthy that the “molecular antibiogram” includes only *bla*_KPC_ in the IRIDICA BAC BSI (Abbott Molecular, Des Plaines, IL, USA) platform—which was discontinued at the time of writing [13]—and only variants of the *bla*_SHV_ and *bla*_CTX-M_ genes encoding for extended spectrum β-lactamases (ESBLs) in the VYOO assay (Analytik Jena, Jena, Germany). Conversely, it is noteworthy that the “molecular antibiogram” in the T2Resistance™ panel—which was under development at the time of writing [13]—includes a relatively comprehensive list of genes encoding for KPC, OXA-48, NDM, VIM, and IMP (carbapenemases), CMY and DHA (AmpC β-lactamases), or CTX-M-14 and CTX-M-15 ESBLs (see also below). In 2019—when our evaluation was conducted—the RUO version of the T2Resistance™ panel did not include CTX-M ESBLs, leaving seven antimicrobial resistance genes evaluable presently. We were unable to assess if any of 10 (or 12) samples that tested KPC-positive with the T2Resistance™ panel (or the BC) testing would also have tested positive for CTX-M. This trait was highly probable based on the epidemiological situation of Gram-negative BSIs in Italy [22]. In view of these observations, the data here presented are preliminary and only partially reflect the diagnostic potential of the T2Resistance™ panel. Including the antimicrobial resistance determinants for Gram-positive organisms, seven groups of genes (*bla*_KPC_, *bla*_CTX-M-14/15_, *bla*_NDM/VIM/IMP_, *bla*_OXA-48_, *vanA*/*vanB*, *mecA*/*mecC*, and *bla*_CMY_/_DHA_) for a total of 13 molecular targets of antimicrobial resistance are currently detectable by the T2Resistance™ panel (https://www.t2biosystems.com/products-technology-ous/pipeline-ous/t2resistance-panel-ous/, accessed on 27 July 2021). Furthermore, there were no positive results for other carbapenemase-encoding genes such as NDM or OXA-48; this was unsurprising considering that our study was a proof-of-concept study completed in a restricted time period. Accordingly, no specific criteria were used for the patient/sample inclusion because we aimed to include the maximum number of single-patient samples as possible. Finally, the panel requires a whole-blood sample of 4 mL (https://www.t2biosystems.com/products-technology-ous/pipeline-ous/t2resistance-panel-ous/, accessed on 27 July 2021), which is consistent with that (1−5 mL) of aforementioned nucleic acid amplification-based tests—in particular the VYOO assay uses a volume of 5 mL of whole blood [6]. If the T2Bacteria^®^ panel or the VYOO assay should be performed in triplicate, volumes of 12−15 mL would be equivalent to the input volume for a single BC—usually at least two sets of BCs per patient are drawn at the time of BSI [8].

## 4. Materials and Methods

Between February 2019 and June 2019, K_2_ EDTA-treated whole blood samples were collected and processed with the T2Resistance™ panel in the clinical microbiology laboratories of two Italian University hospitals (Fondazione Policlinico Universitario A. Gemelli IRCCS of Roma and Ospedale Santa Maria della Misericordia of Perugia). Samples were from patients hospitalized in intensive care units, infectious disease clinics, or hematology wards, who had a laboratory diagnosis of carbapenemase-producing Gram-negative bacterial BSI (see below). The patient/sample selection process, starting from patients who were diagnosed with Gram-negative bacterial BSIs during the study period, is depicted in Figure 2. To obtain a maximum number of samples to test in a five-month period, we did not use a pre-specified window for sample collection. The time elapsing from diagnosis to T2Resistance™ panel sampling/testing ranged from 0.5 h to 57.1 h (i.e., within 3 days), depending on the actual operational hours of the laboratories (e.g., falling on holiday weekends). However, for the majority of samples, this range fell within the time (i.e., 36 h) in which a positive T2MR^®^ based assay result was expected [23].

For each patient, BSI diagnosis due to carbapenemase-producing Gram-negative bacteria relied on direct BC results of the matrix-assisted laser desorption ionization–time of flight mass spectrometry (MALDI-TOF MS) based organism identification followed by the carbapenemase production assessment using the Xpert^®^ Carba-R Test (Cepheid, Sunnyvale, CA, USA) or NG-Test Carba5^®^ (NG Biotech, Guipry, France) assays, which were done according to each manufacturer’s instructions. For confirmation, β-lactamase genes (i.e., *bla*_CMY_, *bla*_KPC_, *bla*_IMP_, *bla*_NDM_, *bla*_OXA-48-like_, *bla*_VIM_, and *bla*_AmpC_) of bacterial isolates obtained from the subcultures of patients’ BCs were PCR sequenced [24,25], and bacterial isolates were subjected to AST using the MICRONAUT broth microdilution panel (Merlin Diagnostika GmbH, Bornheim, Germany) (data not shown). Regarding the T2Resistance™ panel testing, a 4-mL sample from each patient was directly put into the sample inlet, which had first been assembled with the T2Resistance cartridge loaded with the T2Resistance reagent tray. The panel contains all of the disposables and reagents needed to detect antimicrobial resistance genes direct from sample. The assembled panel was then loaded onto the T2Dx, a benchtop, fully automated sample-to-result system that performs all steps of the assay after sample loading. During processing on the T2Dx, an aliquot of the patient sample was directly mixed with DNA amplification reagents. After that, the amplicon was hybridized with target specific probes, which are bound to superparamagnetic particles, and was then detected by T2MR. Finally, the result was accessible on the T2Dx main menu. For each sample, processing was completed according to the T2Dx instructions for use.

## 5. Conclusions

In summary, consistent with the fact that T2MR^®^ technology would detect whole microbial cells after initiation of antimicrobial therapy [23], we believe that the T2Resistance™ panel (combined with the T2Bacteria^®^ panel [18,19]) might be used to diagnose clinically relevant carbapenem-resistant Gram-negative bacterial BSIs. However, further studies are needed to clearly define the T2Resistance™ panel performance characteristics and, importantly, to prove the cost-effectiveness of introducing such a panel in the laboratory diagnostic workflow for BSIs.

## Figures and Tables

**Figure 1 antibiotics-10-00950-f001:**
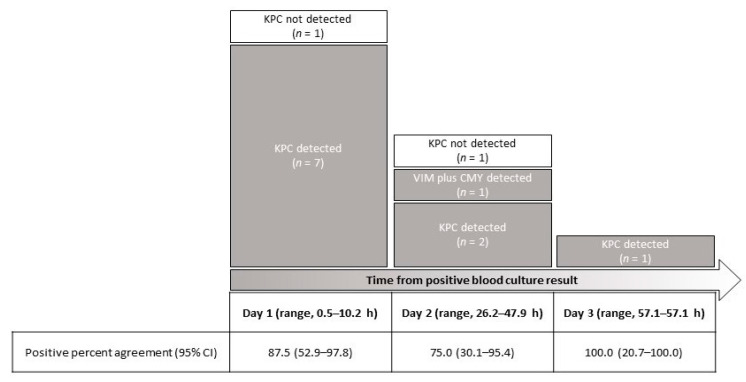
T2Resistance™ panel results for 13 whole blood samples from patients with carbapenemase-producing *Klebsiella pneumoniae* (*n* = 12) or *Citrobacter freundii* (*n* = 1) BSIs stratified according to the time from a positive BC result. Eleven samples were positive for a carbapenemase (10 KPC and 1 VIM) β-lactamase gene, whereas one sample was also positive for an AmpC (CMY) β-lactamase gene.

**Figure 2 antibiotics-10-00950-f002:**
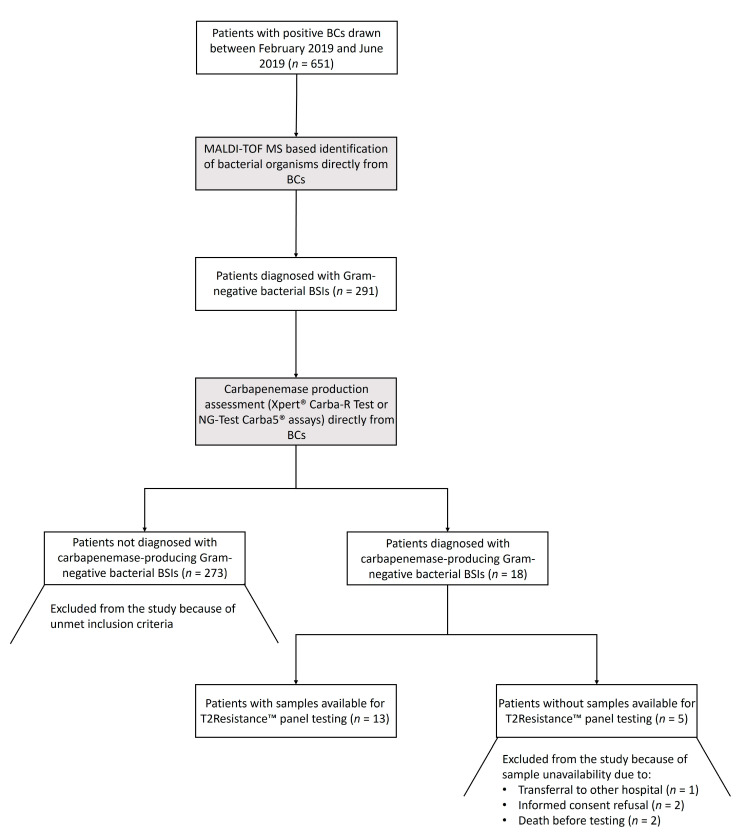
Flow chart for the patient/sample selection process from the study. Patients diagnosed with a carbapenemase-producing Gram-negative bacterial BSI were sampled for testing with the T2Resistance™ panel. Microbiological tests on which the diagnosis was based are highlighted in gray. BCs, blood cultures; MALDI-TOF MS, matrix-assisted laser desorption ionization–time of flight mass spectrometry; BSIs, bloodstream infections.

## Data Availability

The data presented in this study are available on request from the corresponding author.

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
