# Peer review of "Direct Testing for KPC-Mediated Carbapenem Resistance from Blood Samples Using a T2 Magnetic Resonance Based Assay"

_antibiotics, 2021, doi:10.3390/antibiotics10080950_

Round 1

Reviewer 1 Report

Very interesting and promissing approach to the diagnosis of multidrug-resistant bloodstream infections in terms of speed of diagnosis. 

Author Response

Very interesting and promising approach to the diagnosis of multidrug-resistant bloodstream infections in terms of speed of diagnosis.

Answer: We are very grateful to the reviewer for fully appreciating the approach described in this manuscript.

Reviewer 2 Report

The abstract is well written and sounds clear, but I don't understand why the authors have mitigated their conclusion with a more than 80% of positivity rate.

I suggest to the author a thorough bibliography on blaKPC; I found the first paragraph of the introduction a bit shallow. I came across this paper https://www.ncbi.nlm.nih.gov/pmc/articles/PMC6829112/, and these authors show that from nearly half their isolates exhibiting carbapenemase activity, only a few of them carry blaKPC. They suggested that the blaKPC gene is not the main cause of resistance spread within the context of their study. Can you comment on this concerning your finding?

Material and method: The author arbitrary chooses 13 out of 300 samples? I am curious to understand why 13 and why not 40? 50? 100?

Line 182 - 194: definitively, the sample selection procedure does not sound clear to me. Would you please describe a more precise workflow? How many samples with laboratory diagnosis of carbapenemase-producing Gram-negative BSI? Why have you chosen to go by an arbitrary process instead of a funnel flow?

Line 195 - 202: please provide a clear flow chart. I still don’t understand why the author has used MALDI-TOF “OR” Xpert® Carba-R Test? What about the disc diffusion method widely used as a reference?

Author Response

The abstract is well written and sounds clear, but I do not understand why the authors have mitigated their conclusion with a more than 80% of positivity rate.

Answer: We are very grateful to the reviewer for fully appreciating the approach described in this manuscript, as well as for encouraging us to modify our conclusion in the Abstract in order to put further emphasis on the study findings. See page 2, lines 30 to 32 of the revised manuscript.

I suggest to the author a thorough bibliography on blaKPC; I found the first paragraph of the introduction a bit shallow. I came across this paper https://www.ncbi.nlm.nih.gov/pmc/articles/PMC6829112/, and these authors show that from nearly half their isolates exhibiting carbapenemase activity, only a few of them carry blaKPC. They suggested that the blaKPC gene is not the main cause of resistance spread within the context of their study. Can you comment on this concerning your finding?

Answer: As suggested, we expanded the paragraph to highlight the relevance of the “Klebsiella pneumoniae carbapenemase (KPC), which is encoded by the blaKPC gene, as a major determinant of carbapenem resistance among bloodstream isolates from Italy and other European (e.g. Italy, Greece, etc.) or non-European (e.g. Brazil, USA, etc.) countries”. However, we also noted, “other carbapenemase (e.g. NDM) or β-lactamase (e.g. AmpC) encoding genes are increasingly recognized as determinants capable of rendering Gram-negative organisms resistant to multiple antimicrobial drugs”. We supported these sentences with appropriate references. See page 3, lines 48 to 54 of the revised manuscript.

Material and method: The author arbitrary chooses 13 out of 300 samples. I am curious to understand why 13 and why not 40? 50? 100?

Answer: I am sorry for improperly saying “arbitrarily selected”. Thus, we rephrased the sentence as well as we added a new Figure (namely Figure 2) to clarify the patient/sample selection process performed by us. See page 10, lines 196 to 201, and pages 11/12, lines 208 to 212 of the revised manuscript.

Line 182 - 194: definitively, the sample selection procedure does not sound clear to me. Would you please describe a more precise workflow? How many samples with laboratory diagnosis of carbapenemase-producing Gram-negative BSI? Why have you chosen to go by an arbitrary process instead of a funnel flow?

Answer: As stated above, we clarified this issue by adding Figure 2 that depicts how we performed the patient/sample selection process, which actually relied on a funnel flow. See pages 11/12, lines 208 to 212 of the revised manuscript.

Line 195 - 202: please provide a clear flow chart. I still don’t understand why the author has used MALDI-TOF “OR” Xpert® Carba-R Test? What about the disc diffusion method widely used as a reference?

Answer: In addition to clarifying the patient/sample selection process, we modified the sentence regarding the use of both MALDI-TOF MS and Xpert® Carba-R Test, and added details about the phenotypic test (i.e. the broth-microdilution reference method) used to confirm the resistance detection results of the samples studied by us. See page 12, lines 214 to 225 of the revised manuscript.

Reviewer 3 Report

In the present report Authors used T2Resistance™ panel to detect carbapenem resistance genes in 13 infected blood samples. T2Resistance™ panel identified resistance genes in 11 samples and failed to detect them in 2. Because of the low sample size, authors did not perform analyses such as sensitivity and specificity. The sample size in the current study is low to make any definitive claim, but the present study indicates that T2Resistance™ panel can be used to detect the antibiotic resistance markers in blood samples. I have the following minor concerns:

  1. It seems that T2Resistance™ requires 4 mL of the blood sample. If I need to run experiments in triplicate, do I need 12 ml of blood? What about the amount of blood required in other methods?
  2. How many resistance genes can be identified? Can T2Resistance™ be tweaked to profile new resistance genes? Can it be done by the user?
  3. What is the cost and time per sample compared to other commonly used methods?
  4. Failure of detection of resistance gene in two blood samples was attributed to the timing of sampling (2 h after the antibiotic dose, lines 100-102). Please describe when sampling was performed in other cases.

Author Response

In the present report, Authors used T2Resistance™ panel to detect carbapenem resistance genes in 13 infected blood samples. T2Resistance™ panel identified resistance genes in 11 samples and failed to detect them in two. Because of the low sample size, authors did not perform analyses such as sensitivity and specificity. The sample size in the current study is low to make any definitive claim, but the present study indicates that T2Resistance™ panel can be used to detect the antibiotic resistance markers in blood samples.

Answer: We are very grateful to the reviewer for fully appreciating the approach described in this manuscript.

I have the following minor concerns: It seems that T2Resistance™ requires 4 mL of the blood sample. If I need to run experiments in triplicate, do I need 12 ml of blood? What about the amount of blood required in other methods?

Answer: As specified in the paper, the T2Resistance™ panel works with 4 mL of whole blood volume. This is almost equal to the volume required by other nucleic acid amplification-based assays/systems/platforms commercially available today. A comment about these details and the eventual need for larger volumes to run experiments in triplicate was added. See page 9, lines 179 to 185 of the revised manuscript.

How many resistance genes can be identified? Can T2Resistance™ be tweaked to profile new resistance genes? Can it be done by the user?

Answer: The complete list of genes (including those for Gram-positive organisms) detectable by T2Resistance™ panel was added. We believe that the T2Resistance™ panel can be tweaked to profile new resistance genes but that the user cannot do this. See page 9, lines 172 to 175 of the revised manuscript.

What is the cost and time per sample compared to other commonly used methods?

Answer: As required, details about the cost and time per sample were added. See page 8, lines 148 to 153 of the revised manuscript.

Failure of detection of resistance gene in two blood samples was attributed to the timing of sampling (2 h after the antibiotic dose, lines 100-102). Please describe when sampling was performed in other cases.

Answer: As required, details about the timing of sampling in other cases were added. See page 6, lines 106 to 107 of the revised manuscript.

Round 2

Reviewer 1 Report

The revised manuscript should be published

Author Response

Comments and Suggestions for Authors

The revised manuscript should be publish

We are very grateful to the reviewer for fully appreciating the approach described in this manuscript

Reviewer 2 Report

I agree with the majority of the author's reply but sorry I still have a few concerns.

line 29-31: I think the author should avoid "despite", "however" etc.. in the abstract. an abstract is a summary of your finding, not your interpretation/ impression. you can do that in your discussion and/or conclusion.

line 104-109: are you saying that gene amplification can be falsely negative due to anti biotherapy and turned to be culture-positive later?

what is a sample drawing? do you mean sample collection? 

Author Response

I agree with the majority of the author's reply but sorry I still have a few concerns.

Line 29-31: I think the author should avoid "despite", "however" etc.. in the abstract. an abstract is a summary of your finding, not your interpretation/ impression. you can do that in your discussion and/or conclusion.

Answer: As suggested, I deleted this sentence. See page 2, lines 30–31 of the revised manuscript.

Line 104-109: are you saying that gene amplification can be falsely negative due to anti biotherapy and turned to be culture-positive later?

Answer: It is possible provided testing-related technical reasons can be excluded.

What is a sample drawing? do you mean sample collection?

Answer: We changed this to “sampling …” for clarity. See page 6, lines 106–107 of the revised manuscript.

Reviewer 3 Report

I am satisfied with the author’s responses to my questions/issues raised in my initial review and don't have any further concerns.